# Polycyclic Aromatic Hydrocarbon Levels in Wistar Rats Exposed to Ambient Air of Port Harcourt, Nigeria: An Indicator for Tissue Toxicity

**DOI:** 10.3390/ijerph18115699

**Published:** 2021-05-26

**Authors:** Rogers Kanee, Precious Ede, Omosivie Maduka, Golden Owhonda, Eric Aigbogun, Khalaf F. Alsharif, Ahmed H. Qasem, Shadi S. Alkhayyat, Gaber El-Saber Batiha

**Affiliations:** 1Institute of Geo-Science and Space Technology, Rivers State University, P.M.B. 5080, Nigeria; rogers.kanee@ust.edu.ng (R.K.); precious.ede@ust.edu.ng (P.E.); 2Department of Preventive and Social Medicine, Faculty of Clinical Sciences, University of Port Harcourt, P.M.B. 5323, Nigeria; omosivie.maduka@uniport.edu.ng; 3Department of Public Health Services, Rivers State Ministry of Health, Port Harcourt 500001, Nigeria; goldenowhonda@yahoo.com; 4Center for Occupational Health, Safety, & Environment (COHSE), Institute of Petroleum Studies (IPS), University of Port Harcourt, P.M.B. 5323, Nigeria; 5Department of Clinical Laboratory Sciences, College of Applied Medical Sciences, Taif University, Taif 21944, Saudi Arabia; alsharif@tu.edu.sa; 6Laboratory Medicine Department, Faculty of Applied Medical Sciences, Umm Al-Qura University, Mecca 21955, Saudi Arabia; Aaqasem@uqu.edu.sa; 7Department of Internal Medicine, Faculty of Medicine, King Abdulaziz University, Jeddah 21589, Saudi Arabia; dr_shadi2002@hotmail.com; 8Department of Pharmacology and Therapeutics, Faculty of Veterinary Medicine, Damanhour University, Damanhour 22511, AlBeheira, Egypt; dr_gaber_batiha@vetmed.dmu.edu.eg

**Keywords:** particulate matter, polycyclic aromatic hydrocarbons, Port Harcourt metropolis, Wistar rats

## Abstract

This study investigated the PAH levels in Wistar rats exposed to ambient air of the Port Harcourt metropolis. Twenty Wistar rats imported from a nonpolluted city (Enugu) were exposed to both indoor and outdoor air. Following the IACUC regulation, baseline data were obtained from 4 randomly selected rats, while the remaining 16 rats (8 each for indoor and outdoor) were left till day 90. Blood samples were obtained by cardiac puncture, and the PAH levels were determined using Gas Chromatography Flame-Ionization Detector (GC-FID). GraphPad Prism (version 8.0.2) Sidak’s (for multiple data set) and unpaired *t*-tests (for two data sets) were used to evaluate the differences in group means. Seven of the PAHs found in indoor and outdoor rats were absent in baseline rats. The mean concentrations of PAH in indoor and outdoor animals were higher than those of baseline animals, except for Benzo(a)pyrene, which was found in baseline animals but absent in other animal groups. Additionally, Dibenz(a,h)anthracene, Indeno(1,2,3-c,d)pyrene, Pyrene, 2-methyl, and other carcinogenic PAHs were all significantly higher (*p* < 0.05) in outdoor groups. The vulnerable groups in Port Harcourt are at the greatest risk of such pollution. Therefore, urgent environmental and public health measures are necessary to mitigate the looming danger.

## 1. Introduction

Globally, ambient air pollution is a threat to health and the climate, as it is responsible for the majority of recorded mortality and morbidity [1,2]. The urban airborne particulate matter is a significant air quality indicator and a pollutant of public health concern that is responsible for tissue toxicity and high disease burden [1,2,3]. According to WHO’s IARC, soot, petroleum refining, and other industrial exposures and the content’s pollutants are classified as mixtures with health effects of carcinogenic (Group 1) and probable carcinogenicity (Group 2A) [1,4,5] respectively. Airborne particulate matter pollution results from incomplete combustion of organic substances, particularly petroleum derivatives compounds [5,6,7]. Particulate matter contains both organic and inorganic compounds such as; carcinogenic substances such as polycyclic aromatic hydrocarbons (PAHs) and heavy metals [6,7]. PAHs are the most common subclass of polycyclic organic matter and contain only carbon and hydrogen, made up of two or more fused aromatic rings [8]. PAHs with lower molecular weights that have two to three-ring structures are predominantly found in the atmosphere in vapour form, while those with higher molecular weights, four rings, and above are particle-bonded and are said to be more toxic to human health [9,10,11]. These particulate-bonded PAHs environmental pollutants have been found to have low solubility in water and penetrate tissue membranes through hydrophobic attraction, and afterwards, they bio-concentrate in tissues [9,12].

In the body cells, these pollutants accumulate and undergo toxicokinetics and metabolism that result in rapid disadvantageous cellular changes, contributing to acute and chronic morbidities and mortality in the general population, especially the vulnerable individuals relating to age, sex, pregnancy, and routine occupation [13,14,15,16]. The ability of PAHs to travel over long distances, even to very remotes areas, has been reported [4]. Most PAHs are potent carcinogens and are typically attached to particles in the air. Exposure to carcinogenic PAHs occurs primarily in the air via inhalation of particles, and no guideline limit can appropriately be estimated as exposure levels remain relevant to public health risk [16]. Assessment of bioaccumulation of toxic and hazardous compounds in tissues is vital in predicting the health effects and risks of environmental compounds on human health, especially PAHs known for high carcinogenic implications [17]. The WHO IARC and several investigations have implicated polycyclic aromatic hydrocarbons as the major causative factor of cancers, cardiovascular, renal, and respiratory dysfunctions [1,18,19,20].

Unlike places such as Abuja, where relative clean air has been noted [21,22], the ambient air quality of the crude oil-rich city of Port Harcourt has been reported to be polluted with particulate matter of petroleum origin, especially PM_10_ (averaged 494 µg/m3) and PM_2.5_ (averaged 155 µg/m3), and its constituent compounds including PAHs [23,24,25,26]. Using secondary data from hospital records, a significant increase in trends of respiratory morbidity, and mortality in Port Harcourt over the years, have been documented [20,23]. While particulate matter containing PAHs results from incomplete combustion of hydrocarbons, inhabitants are exposed to PAHs through air inhalation and re-suspended soil and dust, consumption of food and water, and dermal contact with soil and dust [10,17,27]. Air inhalation occurs in both indoor and outdoor exposures. While food and water consumption relate mostly to indoor exposures, contact with the skin generally occurs outdoors. Except for certain occupations and social defects, people spent 80–93% of their time indoors, and hence inhalation route would result mostly from indoor air [17,28]. In America and most developed States, food ingestion is likely the largest route of PAH exposure as against inhalation. While drinking-water and soil are generally minor sources of PAHs [17,27], in developing nations, biomass and fossil fuel are generally used for both industrial and domestic combustions, and ambient environmental tobacco smoke (ETS) remains dominant, with no flue and microenvironments fitted with non-airtight; the contribution of inhalation to PAHS exposure could be very high. Hence, inhalation would be the main contributor to the total daily intake of PAHs [17,23,25]. The kinetics, metabolism, and toxicity of PAHs from different routes of exposure have been well demonstrated in published documents [10,29]. Indoor and outdoor inhalation remains of critical discussion, especially in airborne particulate matter polluted ambient, such as the atmosphere of Port Harcourt [23,24,25,26].

The toxicokinetic properties of the individual PAHs differ widely due to differences in physicochemical properties and molecular weight. Upon inhalation into the respiratory pathway, the fate of the PAH is determined by the structure of the PAH, the dimensions, and the chemical nature of the particulate. The PAHs may dissolve from particles and rapidly absorbed from the solutions following biphasic absorption kinetics in the lungs, while the remainder in particles is likely removed through bronchial mucociliary clearance (swallowed), or the PAHs in particles most probably persist in the lungs at long durations [17,30]. The absorption kinetics is said to be dependent on the site of the respiratory tract that the PAHs deposits. It is more rapidly desorbed and absorbed into circulation through type I epithelial cells in the alveolar region [17,30,31], and systemically it is rapidly metabolized [31] as PAHs deposit through the tracheobronchial region, where it is slowly absorbed into the circulation and intensely metabolized locally [17]. Dose-dependent absorption kinetics have also been demonstrated in the lungs of perfused rats at low exposure concentrations, and the absorption of PAHs in the mucosa follows the first-order kinetics with substantial local metabolism. However, at high-exposure doses, the capacity of epithelium to dissolve and metabolize PAHs becomes saturated, and the absorption rate turns zero-order kinetics [17,30,31,32,33]. In the indoor environment, human exposure is likely to follow the low-dose, while first-order kinetics are likely to be observed in ambient air inhalation where concentrations are higher [30,31,32]. PAHs distribute rapidly in the body as lipophilic compounds, which penetrate biological membranes easily [17,30]. Hence, the detectable concentration of PAHs is observable in tissues between a few minutes and hours after exposure, irrespective of the exposure route [17,30,34]. PAHs do not accumulate in the body but are found more in adipose tissues [17]; however, in the lungs, PAHs are not well correlated [35]. PAHs are generally demonstrable in most human tissues, and the risk of toxicity and susceptibility differs across population demographics and tissue sites [17,36]. The metabolism that occurs at the respiratory tract is significant to the toxicity of inhaled particles containing PAHs. Macrophages actively metabolize and engulf particulates containing PAHs in the lungs, transport them to the bronchi and reactive carcinogenic metabolites, which bind covalently to proteins, and nucleic acids are released, resulting in toxicity and carcinogenicity [10,37,38,39]. Three principal pathways are known to activate PAHs for toxic intermediates and further metabolism, including; (dihydro)diol-epoxide formation, radical cation formation, and the o-quinone pathway. Additionally, cascades of enzymes interplay in PAHs’ metabolism, especially CYPs (cytochrome P450s) and epoxide hydrolase. Studies have also found PAHs to cross the placental barrier easily [30,37,40].

Exposure to hazardous compounds such as heavy metals and PAHs from urban airborne particulate matter (PM) pollution inducing tissue toxicity through oxidative stress and mutilation of vascular endothelial cells is the likely cause of the rise in morbidity and mortality of inhabitants in a hydrocarbon-polluted environment [41,42,43]. Depending on the dose and duration of exposure, in vivo and in vitro animal models using rodents and mammal tissues (cultured human and placenta) found that both isolated and or combined PAHs induce and promote skin irritation, inflammation, carcinogenicity, immune suppression, genotoxicity, and mutagenicity and teratogenicity leading to embryotoxic effects [18,43,44,45]. The compounds prompt disease through oxidative stress [46,47] and change in the genome; thus, altering DNA methylation and expression of the specific gene [5,48].

Unfortunately, despite the level of pollution in the South-South region of Nigeria, there have been limited in vitro and or in vivo studies [30] using animal tissues to assess the differential levels of PAHs in indoor and outdoor, and compared to baseline data. This 0study notes that studies using secondary hospital data [20,23], air quality index measurements, and satellite determination of particulate load [24,25,26] reported contradictory findings. This not uncommon as confounding factors other than particulate matter pollution and its constituents could be responsible for the observed change in trends of morbidity. Conversely, aetiology from hospital records is also not reliable because of the lack of data quality and accuracy [49,50,51,52], thus, cannot be linked to environmental pollutants such as particulate matter and PAHs. Though they could show changes for scientific speculations, but cannot give explanations [30]. However, some studies used the Lichen plant as a bioindicator for air pollution in Port Harcourt [53], while other studies used electrical turbines powered by petroleum fuel to generate controlled fumes inhaled by Wistar rats. Others also use intratracheal instillation [30,54]. These studies incorporate designs and method that weakens possible casualty or indicator for toxicity. The lipophilic and hydrophobic toxicokinetic and metabolism in animal tissues [10,17,29] and plant metabolism [53] differ widely. Additionally, humans are not physiologically designed for controlled and episodic fumes inhalation, which is likely to destroy human cells within the inhalation time through carboxyhemoglobin pathogenesis, resulting from carbon monoxide poisoning [55,56] as against normal ambient air [33]. Consequently, simulated findings from such study methods most probably do not correlate relatively to animal health or an indicator for toxicity in human tissues, compared to studies that expose experimental animals (from hydrocarbon and petrochemical pollution-free environment) to the normal atmospheric ambient air accustomed to human respiration [33,54]. To this end, this study, using an animal model [57], compared the PAH level concentrations in the blood tissue of Wistar rats exposed to the indoor and outdoor ambient air of a polluted environment in Port Harcourt and how they deviate from the established baseline obtained from the nonpolluted environment.

## 2. Materials and Methods

### 2.1. Study Design

The animal experimental model study design was used for the study, as shown in Figure 1 below.

### 2.2. Measurement of Indoor and Outdoor PM Concentrations

A Handheld (China Way CW-HAT200) Suspended Particulate Matter (SPM) optical meters were used to monitor the daily (in the mornings and evenings) particulate matter (PM_2.5_ and PM_10_), relative humidity, and ambient temperature in the indoor and outdoor air by an optical method. The measurement was conducted in the mornings (8 am) during exposure of outdoor rats and evenings (5 pm) when they are returned inside. The indoor rats were always inside, and the particulate matter was measured. The daily (90 working days) measurements for both indoor and outdoor air were recorded in a dedicated notebook.

### 2.3. Animal Housing and Care

Twenty Wistar rats (all ≤ 1-month-old, weighing between 25–30 grams) were imported from a non-air-polluted (free of petroleum-based activity) location, the University of Nigeria, Nsukka (Geo-Coordinate: 6.8429 N, 7.3722 E; Altitude: Temp: ~37.6 °C; Relative humidity: 45.9; PM_2.5_; 1–12µg/m^3^ and PM_10_; 4–25µg/m^3^) (Figure 2) to Anatomy animal house, College of Medicine, Rivers State University, Port Harcourt (Geo-Coordinate: Temp ~28 °C; Relative humidity; ~90; PM_2.5_: 32–291.5µg/m^3^ and PM_10_: 33–467 µg/m^3^). The rats were housed in two separate wooden cages with the body made of thin barb-wire (for adequate ventilation) under a controlled environment (room of 12 m height, with double windows and day/night cycle). The water (Eva bottled water, with NAFDAC No. 01–0492) and palette animal feeds (TopFeed animal feeds brand, NAFDAC No. A9–0317) free of PAHs (evaluated via their product content description) were administered to the animals to ensure experimental parameters examined were not re-ingested outside the route (air) being investigated. The rats were made to acclimatize to the animal house for 2 weeks before the commencement of the experiment.

### 2.4. Animal Exposure/Treatment

Following the IACUC regulation [58], 4 randomly selected animals were euthanized following administration of the AGTE on arrival (baseline) and the remaining 16 rats (8 each for indoor and outdoor) were randomly allocated into wooden cages and left for 90 days. While the indoor animals remained inside the room all through the experiment, the outdoor animals were brought out to the open ambient for 8 h (8 am–4 pm) every day and then sent indoors.

### 2.5. Sample Collection

At the end of the 90-day experiment, 3 rats (each at a time) from the respective experimental groups were euthanized using diethyl ether in a desiccator. After 2 min, the rats (each at a time) were dissected, and 4 mL of blood collected through cardiac puncture.

### 2.6. Analyte Extraction and Sample Preparation

The PAHs were extracted using a liquid–liquid extraction process. This was done by adding 20 mL of Dichloromethane (DCM) to 4 mL of the blood sample. The extractant was added in the vials, capped, and vortexed in the centrifuged set at 300 r.p.m for 20 s. The organic layer was separated using a pipette attached with a pipette filter into a thermally treated and cleaned amber glass bottle. Silica gel fractionation was done to clean up the samples for the analyte to be eluted. The eluted samples were then concentrated by nitrogen blow-down and transferred into a GC vial for analysis. 

### 2.7. GC-FID Analysis

The reconstituted, clean extracted samples were analysed at Analytical Concept LTD Poultry Rd. by 2nd Railway, Odani Green City, Elelenwo Port Harcourt, Rivers State, Nigeria. The total PAHs and the individual PAH concentrations were determined using gas chromatography (GC) equipped with a Flame-Ionization Detector (FID). The peaks were identified using the GC retention time for each PAH in the analyte sample. The blood-sample-extracted analyte was analysed using an HP 5890 series II Gas Chromatography Flame-Ionization Detector equipped with an HP-1 (Methyl Silicone Gum) Capillary column (Agilent, 30 m × 0.32 mm × 0.00025 mm film thickness) operating with Helium as carrier gas at 2 µL per mL. The injection volume was 2 µL, using a splitless inlet mode. The GC temperature program was set according to an established method [36] with slight modification. The oven was configured to 60 °C and held for 1 min, ramped to 320 °C at the rate of 9 °C/min, and held for 5 min. The injector and the FID were held at 275 °C and 325 °C, respectively. The solvent’s (in the solution used in extraction and storage) peak was first noted in the GC-FID screen. Afterward, the spikes of the lightest compounds in the PAHs to the heaviest and their respective concentrations were detected, and the total PAHs were recorded. The retention time (from injection to detection) of the GC-FID was 40 min.

### 2.8. Quality Assurance and Quality Control

First, the Wistar rats were imported from a non-petroleum industrial environment (Nsukka, Enugu State) of about 233 KM away from Port Harcourt (see Figure 2) to minimize the presence of PAHs and particulate matter other than petrogenic and petrochemical sources. During sample preparation and preservation, possible coagulation of the collected blood samples was prevented by the use of a heparinized tube for onward storage, preservation, and centrifugation. 

PAHs are known to be sensitive to UV light, and to prevent its damage, the plasma was instantly transferred to an amber bottle. Before using the GC-FID for analysis, the instrument was left on for one hour to heat the system and limit the availability of unwanted compounds inside the instrument. To further clean and clears the GC’s column, blank organic solvent (DCM) was injected twice into the column before the actual analyte sample and twice after the analysis. These were done to prevent possible interference on the investigated samples by unwanted compounds from carry-over from the previous analysis. Sample IDs were inserted and analysis type (PAH) selected for the instrument to get ready. Each sampler was calibrated before and after each sampling event. Calibration curves were prepared for each PAHs investigated using GC output signal responses (shown by the peak area in the screen chromatogram, from lightest compound to heaviest) at a specific retention time (40 min in total) against the concentration of the analyte injected into the GC through an injection pot. Validation ions and retention times were also used to ensure the identification of the analytes. The detection was 0.00 mg per gram. 

For recovery analysis, to ensure an adequate quantification, the metric spike method was used. The compound was spiked into separately prepared aliquots of the MM5 train condensate samples before analysis. The spiked aliquots are then analysed, and the spike recovery is calculated. The recovery of these spikes (at ≥80%) provided an independent indicator of method accuracy and quantification relative to the sample matrix by assuming that the spiking compound has chemical characteristics that are identical to the PAH target compounds. 

### 2.9. Statistical Analysis

The animal experimental data were analysed using GraphPad Prism (version 8.0.2) software manufactured by GraphPad Software Inc. (San Diego, CA, USA). The data were summarized as mean (±S.D), and inferential statistics were used to test the level of significance (at a 95% confidence level) difference in the mean parameters. Sidak’s multiple comparison test was used to evaluate the differences in the three groups, while Student’s *t*-test was used when only two groups were available.

## 3. Results

The daily indoor and outdoor particulate matter levels were recorded within the ranges of 32 µg/m^3^ to 467 µg/m^3^ (PM_2.5_: indoor = 32–291.5µg/m^3^ [median; indoor = 48.75, outdoor = 79.75] and PM_10_: 67.5–467µg/m^3^ [median; indoor = 114.25, outdoor = 150.00]).

The mean concentrations of individual PAHs in the respective animal groups are presented in Table 1. When they were undetected in the animal group, it was stated as N/F. The line graph distributions of the PAH levels are presented in Figure 3, and the test of mean differences in the PAHs concentrations in the animal groups are presented in Figure 4, Figure 5, Figure 6, Figure 7, Figure 8, Figure 9 and Figure 10.

In baseline animals, some PAHs were undetected. The mean concentrations of Benz(a)anthracene, Benzo(b)fluoranthene, Benzo(k)fluoranthene, Fluorene, Chrysene, and Phenanthrene were not detected in the baseline animal groups. Benzo(a)pyrene B[a]P, a major PAHs carcinogenic maker, was detected with a mean concentration of 1.58×10^−4^ ± 1.55×10^−5^. The mean concentrations of 2-methyl Naphthalene, Acenaphthene, Acenaphthylene, and Anthracene were observed to be 8.78 × 10^−4^ ± 2.00 × 10^−5^, 3.85 × 10^−5^ ± 1.56 × 10^−6^, 1.25 × 10^−4^ ± 7.02 × 10^−6^, and 2.82 × 10^−4^ ± 3.70 × 10^−5^, respectively. While those of Dibenz(a,h)anthracene, Indeno(1,2,3-c,d)pyrene, and Pyrene were noted as 6.12 × 10^−4^ ± 1.30 × 10^−5^, 2.22 × 10^−4^ ± 1.86 × 10^−5^, and 3.94 × 10^−4^ ± 1.80 × 10^−5^, respectively. In the indoor animals, the mean concentrations of 2-methyl Naphthalene, Acenaphthylene, and B[a]P were not detected. While the mean concentrations of Acenaphthene, Anthracene, Benz(a)anthracene, Benzo(b)fluoranthene, Benzo(b)fluoranthene, and Chrysene were 6.99 × 10^−5^ ± 4.16 × 10^−6^, 4.62 × 10^−4^ ± 4.00 × 10^−5^, 4.57 × 10^−5^ ± 2.47 × 10^−6^, 6.84 × 10^−4^ ± 4.97 × 10^−4^, 1.89 × 10^−3^ ± 1.51 × 10^−4^, and 4.04 × 10^−3^ ± 2.26 × 10^−3^, respectively, and those of Dibenz(a,h)anthracene, Fluoranthene, Fluorene, Indeno(1,2,3-c,d)pyrene, Phenanthrene, and Pyrene were found to be, respectively, 1.90 × 10^−5^ ± 1.05 × 10^−4^, 6.83 × 10^−5^ ± 2.35 × 10^−6^, 1.69 × 10^−4^ ± 1.47 × 10^−5^, 3.08 × 10^−3^ ± 1.06 × 10^−4^, 3.16 × 10^−5^ ± 2.37 × 10^−6^, and 7.06 × 10^−4^ ± 1.19 × 10^−5^. In the outdoor animals, B[a]P and Fluoranthene were not detected. The mean concentrations of 2-methyl Naphthalene, Acenaphthene, Acenaphthylene, Anthracene, Benz(a)anthracene, Benzo(b)fluoranthene, and Benzo(k)fluoranthene were noted to be 9.71 × 10^−4^ ± 5.16 × 10^−5^, 8.18 × 10^−4^ ± 2.10 × 10^−5^, 2.30 × 10^−4^ ± 2.06 × 10^−5^, 1.28 × 10^−3^ ± 1.29 × 10^−4^, 8.15 × 10^−5^ ± 2.66 × 10^−6^, 2.93 × 10^−3^ ± 2.37 × 10^−4^, and 2.89 × 10^−3^ ± 1.17 × 10^−4^, respectively. While those of Chrysene, Dibenz(a,h)anthracene, Fluorene, Indeno(1,2,3-c,d)pyrene, Phenanthrene, and Pyrene had mean concentrations of 8.66 × 10^−3^ ± 4.50 × 10^−4^, 2.08 × 10^−2^ ± 1.86 × 10^−3^, 1.87 × 10^−4^ ± 2.93 × 10^−5^, 1.42 × 10^−2^ ± 7.00 × 10^−4^, 4.74 × 10^−5^ ± 4.29 × 10^−6^, and 5.22 × 10^−3^ ± 1.88 × 10^−4^, respectively (Table 1). The graphical distribution showed that Dibenz(a,h)anthracene and Indeno(1,2,3-c,d)pyrene had the highest mean concentrations noted in the outdoor animal group (Figure 3).

Figure 4, Figure 5, Figure 6, Figure 7, Figure 8, Figure 9 and Figure 10 represent the differences in the baseline, indoor, and outdoor PAHS concentrations in tissues of experimental animal groups. The outdoor animals had significantly higher mean values for both 2-methyl Naphthalene (*p* = 0.043) and Acenaphthylene (*p* < 0.0011) relative to the baseline (Figure 4A and Figure 5A). The highest concentrations of Acenaphthene and Anthracene were observed in outdoor animals, which were significantly greater than the levels in the indoor (*p* < 0.0001) and baseline (*p* < 0.0001) animals; however, the differences in the Acenaphthene (*p* = 0.061) and Anthracene (*p* = 0.098) levels in baseline and outdoor animals were not significant (Figure 4B and Figure 5B).

The detected levels of Benz(a)anthracene and Benzo(b)fluoranthene in Figure 6A,B remained significantly higher in the outdoor (Benz[a]anthracene; *p* = 0.043 and Benzo[b]fluoranthene; *p* = 0.002), while it was undetected in the baseline. Similarly, Benzo(k)fluoranthene (*p* = 0.0008) and Chrysene (*p* = 0.026) levels were significantly higher in the outdoor animals when compared to the indoor animals (Figure 7A,B).

In Figure 8A, the difference in the mean Dibenz(a,h)anthracene concentrations of baseline and indoor animals was not significant (*p* = 0.471); however, the outdoor was significantly higher than the baseline (*p* < 0.0001) and indoor (*p* < 0.0001). Fluoranthene was undetected in the baseline animals, but the detected levels in indoor and outdoor animals were not significant (*p* = 0.380) (Figure 9). The mean Pyrene levels in all animals were significantly different (*p* < 0.05), with significantly higher values observed for outdoor when compared with the baseline and indoor animals (*p* < 0.0001). The indoor animals also had significantly greater mean concentrations than the baseline (*p* = 0.038) (Figure 10).

## 4. Discussion

The World Health Organization’s International Agency for Research on Cancer (IARC) established that outdoor air pollution is carcinogenic to humans, as the association between increased incidence of cancer and particulate matter has been established [59]. The presence of daily PM_2.5_ at 32–158 µg/m^3^ and PM_10_ at 33–467 µg/m^3^, which is beyond WHO acceptable limits, confirms the likelihood of Port Harcourt residents being exposed to hazardous compounds [60,61]. This study was triggered by the observed increase in the particulate matter, which is suggested as the likely cause of the relative upsurge in trends of morbidity and mortality recorded over the years in Port Harcourt [19,20,62,63,64]. There are several epidemiological studies suggesting that tissue toxicity from exposure to particulate matter bonded with PAHs are associated with mortality, cancers, upsurge in respiratory, cardiovascular, and renal dysfunctions [43,45,65].

High amounts of fine particulate matter in indoor and outdoor environs noted in the study were similar to those found in highly petroleum-based industrial areas [23,24,25,26,27,62,63,64]. Studies have implicated infants and children being at risk of fine aerodynamic particulates [22,66]. The study findings also aggress with earlier reports of weight loss and skin toxicity as observed on the outdoor experimental rats due to exposure to particulate matter and effects of PAHs [42,67]. Over 500 PAHs and its related compounds have been found in the air [10] and the carcinogenic PAHs such as Benz[a]anthracene, Benzo[a]pyrene, Benzo[b]fluoranthene, Benzo[k]fluoranthene, Chrysene, Dibenzo[a, h]anthracene, Benzo[ghi]perylene, and Indeno[1,2,3-cd]pyrene [10,28]. Most of the PAHs found in the study were mostly of the carcinogenic class. There is strong evidence that injuries to the lung tissues and DNA damage could result from sub-chronic exposure to low amounts of particles, even below the WHO’s 24-h-threshold [33].

Five established carcinogenic PAHs: Benz[a]anthracene, Benzo[b]fluoranthene, Benzo[k]fluoranthene, Chrysene, Benzo[ghi]perylene, and Indeno[1,2,3-cd]pyrene [5,23], including Phenanthrene, were not found in samples from the baseline animal group as against those of indoor and outdoor animal groups (see Table 1). This suggests that people confined to indoor activities are likely to accumulate PAHs and general tissue toxicity compared to residents inhaling air polluted with PAHs bonded particulate matter due to petroleum and hydrocarbon-related industrial activities in the ambient air [61,62,63,64,65,66,67]. Additionally, the continuous inhalation of PAHs-bounded particulate increases the concentrations of these compounds in the tissues, enabling toxicity through oxidative stress [46,47], inflammation, and DNA methylation [5,48].

Nsukka is located in Enugu, south-east Nigeria. It is an agricultural town, with the University of Nigeria (UNN) situated in it and having minimal commercial and industrial activities that are likely to generate pollutants such as PAHs [68,69]. As seen in Figure 2 above, the experimental animals were imported from this non-petroleum and hydrocarbon pollution-free environment (UNN) that should ideally record no concentrations of PAHs. Although in lesser amounts, the study found some concentrations of the individual PAHs, including B[a]P in blood samples of the baseline animal group. This is likely to result from pyrogenic sources, including domestic combustion, and most probably not from hydrocarbon or petrogenic events [36,70,71]. This calls for a further forensic investigation of the presence of abundant alkyl homologs in such an environment.

Though absent in the baseline animal group, the presence of fluoranthene in the ambient of the indoor animal group (see Table 1) indicates environment-associated tissue bioaccumulation that can induce toxicity, as fluoranthene is said to be a complementary indicator to B[a]P [5]. Studies have demonstrated the burning of biomass, fossil fuel, and solid fuel as important sources of airborne particulates and their component pollutants, including PAHs, which are emitted into indoor environs through unvented or flueless combustion [71,72]. Reports have shown that low-molecule (less than for rings) PAHs predominantly concentrates indoor air [73]; thus, the high concentrations of large-molecule PAHs (including Dibenz[a,h]anthracene, Benzo[b]fluoranthene, Benzo[k]fluoranthene, and Indeno[1,2,3-cd]pyrene), known for high carcinogenicity and toxicity noted in the indoor animal groups compared to those of baseline, suggests intrusion of outdoor air [5,74]. Cooking activities and indoor combustion sources, such as fossil fuels and biofuel [75], and indoor environments impacted by ETS [10] have been reported. Suggesting a higher concentration of some PAHs in indoor exposures likely to occur during the cooking period [17,75]. In as much as outdoor air influences the concentrations of PAHs indoors, the type of cooking fuel used impacts greatly on indoor PAH levels in descending order of dung cake > dung cake/wood mixture > wood > coal > kerosene > LPG [5,17,75]. Additionally, the cooking temperature has been found to influences the production of most indoor PAHs as increasing temperatures affects the evaporation of PAHs into the air and pyrolysis from partially cracked organic compounds [17].

Higher indoor concentrations of 12 PAHs (Naphthalene, Acenaphthene, Fluorene, Phenanthrene, Anthracene, Fluoranthene, Pyrene, Benz[a]anthracene, Chrysene, Benzo[k]fluoranthene, Benzo[e]pyrene, and Benzo[a] pyrene) have been recorded across homes with an unvented fireplace, lacking airtight stoves and windows, homes and public places (such as restaurants, discotheques, and clubs) with ETS, while lower levels were noted across offices, hospitals, schools, libraries, and coffee shops [17,76,77,78,79,80,81] and children were noted to be adversely affected [82]. Conducting the study during the dry season could also have influenced the increased PAH concentration in the ambient. Though seasonal variations were not investigated, earlier reports have established seasonal differences in PAH concentrations, with higher levels in rainy (winter) than in dry or summer seasons [73]. Thus, outdoor air, indoor combustion sources, cooking activities, cooking temperatures, indoor ETS, ventilation characteristics of the living house, climatic factors, and seasonal variations all affect indoor PAH concentrations. This implicates women, especially in Africa, have a higher chance of being affected by indoor air pollutants such as PAHs [80] as their activities are observed to be predominantly indoors.

High amounts of PAHs in the outdoor air most probably indicates higher toxicity of ambient air inhaled by residents and implicates the findings of the previous study that used lichen as a bioindicator for air pollution [53]. The combustion in energy and transformation industries, non-industrial combustion plants, combustion in the manufacturing industry, production processes, traffic road transport, ETS, other mobile sources, waste incineration, agriculture and forestry, and natural sources were all found to contribute significantly to outdoor PAH concentrations in industrialized nations with the variety of sources and emission factor across the countries [83,84]. High toxicity was predicted for pregnant women in these studies [83]. In developing countries, pyrogenic and petrogenic combustions for industrial, commercial, and domestic uses involving biomass, fossil fuel, and solid fuel, including petroleum and hydrocarbon refining sources, contribute to outdoor particulate matter containing PAHs [71,72]. Although no empirical emission inventory has been conducted for Port Harcourt, places with petrochemical economic and industrial ecology have been reported to have ambient air largely polluted by petroleum and hydrocarbon [23,24,25,26,62,71,85,86], which is most probably responsible for the increase in trends of morbidity and mortality [20,23] noted in the metropolis.

With outdoor animal group recording higher concentrations of total PAHs (see Figure 3 and Figure 4), especially the larger molecule, including Dibenz[a,h]anthracene, Benzo[b]fluoranthene, Benzo[k]fluoranthene, and Indeno[1,2,3-cd]pyrene, and the recorded experimental dead Wistar rats notably from the outdoor group, the continuous inhalation of ambient air in Port Harcourt suggests higher chances of tissue toxicity. In particular, inhabitants engaged in outdoor activities, such as road sweepers, street hawkers, traffic wardens, and building and road constructors, are at increased risk. This could be responsible for the clinician’s hematological findings of rare toxins in blood samples of some young patients at tertiary hospitals in Port Harcourt [23]. Hence the medical advice to residents to avoid places where crude oil is cooked, as management of neoplasms and myelofibrosis conditions resulting from crude oil pollution remains expensive, especially for low-income people [23].

From the study result of outdoor PAHs concentration, it is most probable that the indoor air conditions were influenced by outdoor air [5,74]. This might be further worsened by ETS at open spaces adjacent to night clubs. Unlike to the risk of indoor air PAHs concentration, men in Africa are highly at risk of PAHs concentration in outdoor air environs [86] as they carry out more outdoor occupations. Ambient air in densely populated and metropolitan locations has been demonstrated to have higher PAH concentrations than less dense, rural, forest, and agricultural areas because of the numerous sources of fossil fuel combustion [27]. Carcinogenic PAHs, such as Dibenzo[a, h]anthracene, Benzo[ghi]perylene, Indeno[1,2,3-cd]pyrene, and Pyrene, had the predominant concentrations of PAHs in the study. This agrees with previous studies in which toxicity resulting from ambient exposure in petroleum industrial places was demonstrated [23,62,85].

While the levels of the PAH, especially Benz(a)anthracene, Benzo(b)fluoranthene, and Fluoranthene, remained significantly higher in the outdoor animal group, they were often undetected in the baseline animals. This suggests that inhabitants of Port Harcourt metropolis are inhaling ambient air poisoned with PAHs due to petroleum activities into their body systems. Figure 3 and Figure 4 also implicatively show that outdoor residents of Port Harcourt are more predisposed to morbidity and mortality associated with PAHs [19,38,42,87,88,89] unlike those confined indoor [54]. This could explain hematologists’ sudden notice of myelofibrosis in some young residents of Port Harcourt involved in artisanal petroleum refining [23]. The clinician observation negates empirical evidence of myelofibrosis found to be common in older people of 60+ years [23,90,91]. Studies by the United Nations Environmental Programme have shown that substandard oil explorations and artisanal petroleum refining significantly added to the effect of hydrocarbon pollution, which affects both the environment and health [92]. Adversely, exposure to PAHs poses dangers to residents of Port Harcourt, as continuous exposure to outdoor and indoor air polluted with PAHs from petroleum industrial activities is likely to increase cases of cancers [59,62,93] and deformities, especially in pregnant women, infants and children, and obese persons [20,62]. Adverse birth outcomes, such as preterm birth, early pregnancy fetal death, and low birth weight are likely to be common, as well as intrauterine growth retardation with the capacity to distort the academic performance of children in the future when compared to outdoor dominate kids [19,92,93,94].

Unlike studies where both indoor and outdoor PAHs concentrations were observed to be similar [95], the PAH levels found in this study were significantly higher in the outdoor animals when compared to the indoor animals, as shown in Figure 3, and the mean Pyrene levels in all of the animals were significantly different, with significantly higher values observed for outdoor animals (Figure 3 and Figure 4). These findings implicate exposure to the variation of doses of PAHs [96], suggesting that when compared to indoor workers, outdoor workers, such as street traders, beggars, market sellers, road construction workers, and children in open schools within the Port Harcourt metropolis, are exposed to the adverse health effects associated with PAHs induced by petroleum-based particulate matter pollution [16,18]. The notable negation of study results from similar studies [97,98,99] in which higher levels of indoor PAHs were observed is likely a result of the difference in geographic location, seasonal variations, and source of particulate matter inducing the PAHs [73,76]. Deaths recorded from the outdoor experimental exposed rats are likely the result of oxidative stress [46,47] and tissue toxicity beyond the bearable limit and immune suppression prompted by oxidative stress and alteration of DNA methylation and expression of specific genes [43,65,95,96,100,101,102,103], as caution was taken to prevent physical stress and trauma on the rats. The study findings explain and implicate previous reports on increased trends of selected mortality and morbidity associated with particulate matter pollution in Port Harcourt between 2016 and 2018, as well as an upsurge in health burden [20,23]. Several studies found that people living in areas with high concentrations of PAHs, such as hydrocarbon industrial polluted environments, are at a higher risk of toxicity, carcinogenicity, and mutagenicity [5,43,54,62,85,103,104,105] than those in non-hydrocarbon industrial areas, such as Enugu [69].

The presence of carcinogenic PAHs, such as Dibenzo[a, h]anthracene, Benzo[ghi]perylene, Indeno[1,2,3-cd]pyrene, and Pyrene, in the animals indicate the likelihood of tissue toxicity of exposed inhabitants [17,23]. The major concern is the exposure, duration, dose, and bioaccumulation. Toxicity resulting in high incidences of deaths and cancers of several types and at different tissue locations have been documented in places with exposure to carcinogenic PAHs in petroleum industrial ecology [17,62,106,107,108,109,110]. The death of 7 outdoor experimental rats before the end of the 90-day study period, loss of weight, and a change in the skin colour of the exposed animals noted in the study remain a concern, and the likelihood of such effects on human inhabitants is most probable [17,23,62,108,110]. Additionally, the interactions of PAHs with tissues have been found to result in cancers and mortality [110]. Animal model studies have demonstrated lymphoreticular system tumours associated with exposure to Dibenzo[a, h]anthracene, Benzo[ghi]perylene, and Indeno[1,2,3-cd]pyrene [111,112,113]. Implicatively, this suggests that exposed inhabitants were most probably susceptible to the toxicity of PAHs [114,115,116,117,118,119,120,121,122,123,124,125,126,127,128,129,130,131,132,133,134,135] due to its suppression of immunosurveillance and immunocompetence of the body [136,137,138,139,140,141].

## 5. Conclusions

The findings demonstrate that the PAH level in the atmosphere of Port Harcourt metropolis is notably high, unlike places such as Enugu [69], which indicates a high probability of accumulation and toxicity in human tissues and health. Unlike inhabitants in non-hydrocarbon and petrochemical-free environments [69], continuous inhalation of air polluted with toxic particulate matter and high concentrations of PAH in petroleum refining places such as Port Harcourt could exacerbate the already high prevalence of morbidity and health burden associated with airborne particulate matter pollution. With a continuous rise in PAH levels, the risk of chronic diseases such as cancers, respiratory cardiopulmonary dysfunctions, kidney, and other systemic failure and mortality remains most probably high in no distant time. A surveillance framework is instituted for clinicians and laboratory experts’ prompt identification of rare diseases such as myelofibrosis, birth defects, and others. Outdoor workers, especially those involved in petroleum refining, road sweepers, traffic managers, constructions work, infants and children, pregnant mothers, obese persons as well as the elderly are more at risk of toxicity, morbidity, and mortality associated with the diverse PAH concentrations in the ambient air. Systematic emission inventory and routine air quality monitoring are necessary for highly polluted hydrocarbon industrial places to enhance the detection of indoor and outdoor sources of airborne particulates. There are strong shreds of evidence suggesting PAHs are associated with injuries to the lung tissues, DNA damage, and neurologic disorders, resulting from sub-chronic exposure to a low amount of particulate matter and PAH concentrations, even below the WHO’s 24hours threshold [17,33,36,103,104,105]. While the control and elimination of emission sources airborne particulates are strongly recommended, there is an urgent need for appropriate public health screenings for carcinogenicity, nutritional therapy, and environmental interventions to control the potential effects of PAHs and concentrations of carcinogenic compounds in the ambient air of Port Harcourt.

## Figures and Tables

**Figure 1 ijerph-18-05699-f001:**
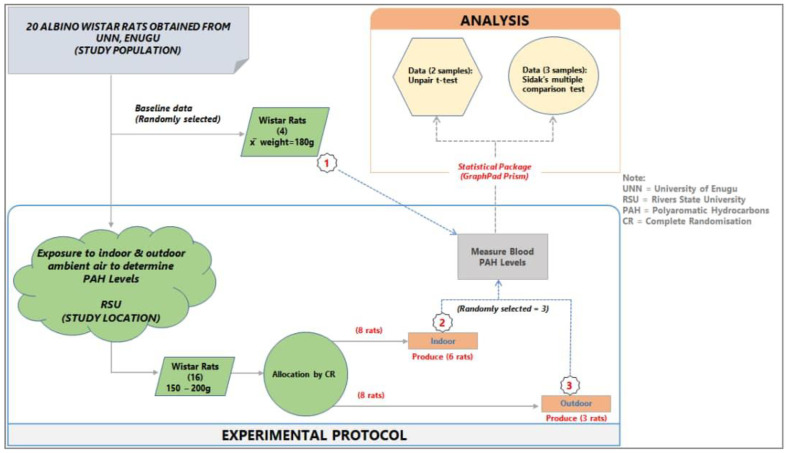
Study design and experimental protocol.

**Figure 2 ijerph-18-05699-f002:**
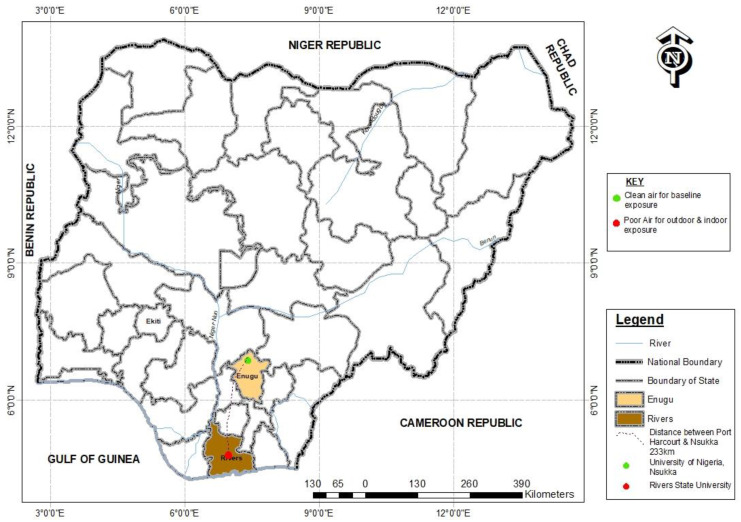
Study areas.

**Figure 3 ijerph-18-05699-f003:**
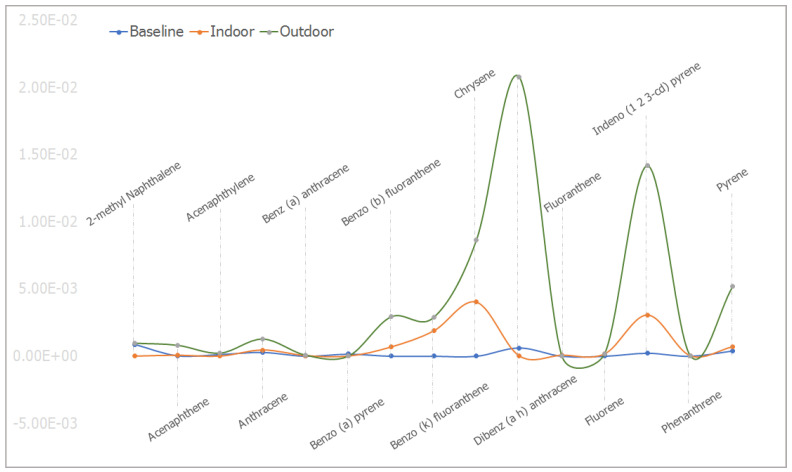
The mean concentrations (in ppm) for the individual PAHs in baseline, indoor, and outdoor experimental animal groups.

**Figure 4 ijerph-18-05699-f004:**
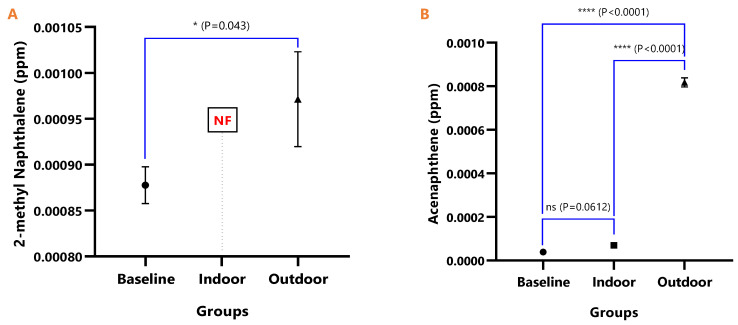
The mean concentrations and test of mean difference for (**A**) 2-methyl Naphthalene and (**B**) Acenaphthene in the various animal groups.

**Figure 5 ijerph-18-05699-f005:**
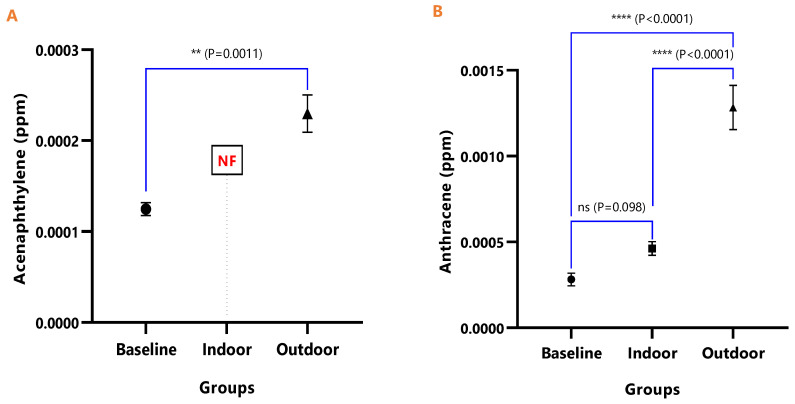
The mean concentrations and test of mean difference for (**A**) Acenaphthylene and (**B**) Anthracene in the various animal groups.

**Figure 6 ijerph-18-05699-f006:**
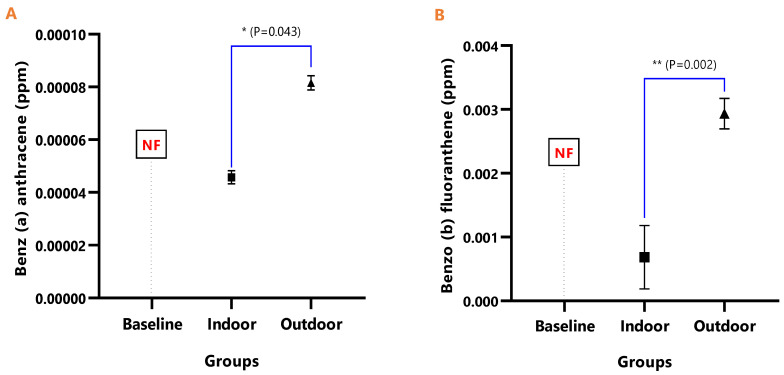
The mean concentrations and test of mean difference for (**A**) Benz (a) anthracene and (**B**) Benzo (b) fluoranthene in the various animal groups.

**Figure 7 ijerph-18-05699-f007:**
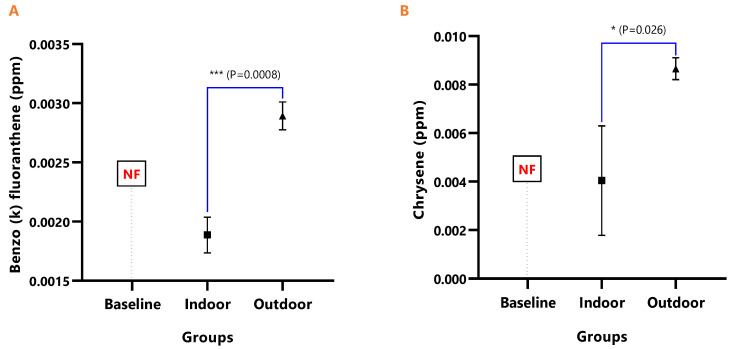
The mean concentrations and test of mean difference for (**A**) Benzo (k) fluoranthene and (**B**) Chrysene in the various animal groups.

**Figure 8 ijerph-18-05699-f008:**
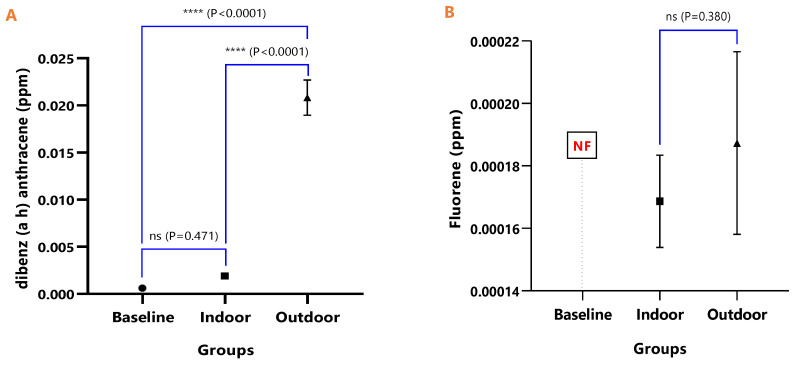
The mean concentrations and test of mean difference for (**A**) Dibenz (a h) anthracene and (**B**) Fluoranthene in the various animal groups.

**Figure 9 ijerph-18-05699-f009:**
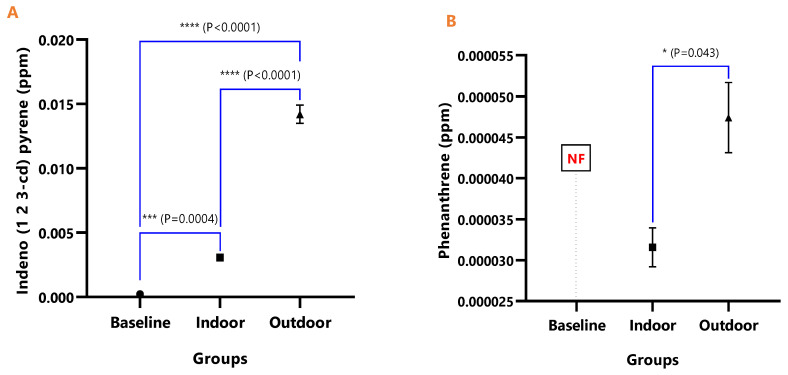
The mean concentrations and test of mean difference for (**A**) Indeno (1 2 3-cd) pyrene and (**B**) Phenanthrene in the various animal groups.

**Figure 10 ijerph-18-05699-f010:**
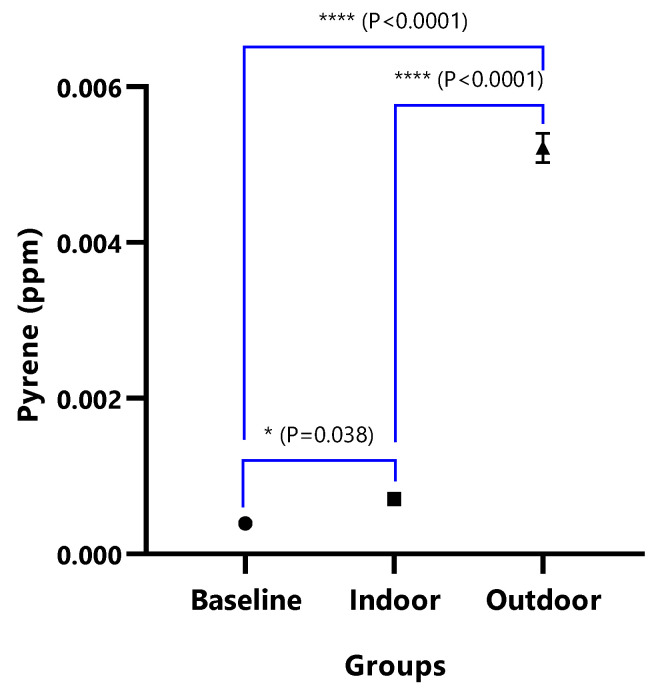
The mean concentrations and test of mean difference for Pyrene.

**Table 1 ijerph-18-05699-t001:** The mean(±S.D) concentrations of PAHs (U/L) in baseline, indoor, and outdoor experimental animal groups.

Polyaromatic Hydrocarbons (PAHs) in ppm	Animal Grouping (Values in Mean ± S.D)		
	Baseline	Indoor	Outdoor
2-methyl Naphthalene (ppm)	8.78 × 10^−4^ ± 2.00 × 10^−5^	N/F	9.71 × 10^−4^ ± 5.16 × 10^−5^
Acenaphthene (ppm)	3.85 × 10^−5^ ± 1.56 × 10^−6^	6.99 × 10^−5^ ± 4.16 × 10^−6^	8.18 × 10^−4^ ± 2.10 × 10^−5^
Acenaphthylene (ppm)	1.25 × 10^−4^ ± 7.02 × 10^−6^	N/F	2.30 × 10^−4^ ± 2.06 × 10^−5^
Anthracene (ppm)	2.82 × 10^−4^ ± 3.70 × 10^−5^	4.62 × 10^−4^ ± 4.00 × 10^−5^	1.28 × 10^−3^ ± 1.29 × 10^−4^
Benz(a)anthracene (ppm)	N/F	4.57 × 10^−5^ ± 2.47 × 10^−6^	8.15 × 10^−5^ ± 2.66 × 10^−6^
Benzo(a)pyrene (ppm)	1.58 × 10^−4^ ± 1.55 × 10^−5^	N/F	N/F
Benzo(b)fluoranthene (ppm)	N/F	6.84 × 10^−4^ ± 4.97 × 10^−4^	2.93 × 10^−3^ ± 2.37 × 10^−4^
Benzo(k)fluoranthene (ppm)	N/F	1.89 × 10^−3^ ± 1.51 × 10^−4^	2.89 × 10^−3^ ± 1.17 × 10^−4^
Chrysene (ppm)	N/F	4.04 × 10^−3^ ± 2.26 × 10^−3^	8.66 × 10^−3^ ± 4.50 × 10^−4^
Dibenz(a,h)anthracene (ppm)	6.12 × 10^−4^ ± 1.30 × 10^−5^	1.90 × 10^−5^ ± 1.05 × 10^−4^	2.08 × 10^−2^ ± 1.86 × 10^−3^
Fluoranthene (ppm)	N/F	6.83 × 10^−5^ ± 2.35 × 10^−6^	N/F
Fluorene (ppm)	N/F	1.69 × 10^−4^ ± 1.47 × 10^−5^	1.87 × 10^−4^ ± 2.93 × 10^−5^
Indeno(1,2,3-c,d)pyrene (ppm)	2.22 × 10^−4^ ± 1.86 × 10^−5^	3.08 × 10^−3^ ± 1.06 × 10^−4^	1.42 × 10^−2^ ± 7.00 × 10^−4^
Phenanthrene (ppm)	N/F	3.16 × 10^−5^ ± 2.37 × 10^−6^	4.74 × 10^−5^ ± 4.29 × 10^−6^
Pyrene (ppm)	3.94 × 10^−4^ ± 1.80 × 10^−5^	7.06 × 10^−4^ ± 1.19 × 10^−5^	5.22 × 10^−3^ ± 1.88 × 10^−4^
Total (ppm)	1.2 × 10^−3^ ± 1.31 × 10^−4^	3.27 × 10^−3^ ± 3.2 × 10^−3^	2.07 × 10^−2^ ± 3.81 × 10^−3^

**Note:** ppm, parts per million; S.D, standard deviation; N/F, not found.

## Data Availability

The data used in this study can be assessed from Harvard dataverse repository at https://doi.org/10.7910/DVN/4GKFHE (accessed on 16 June 2020).

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
