# Peer review of "Polycyclic Aromatic Hydrocarbon Levels in Wistar Rats Exposed to Ambient Air of Port Harcourt, Nigeria: An Indicator for Tissue Toxicity"

_ijerph, 2021, doi:10.3390/ijerph18115699_

Round 1

Reviewer 1 Report

This manuscript investigated the PAH levels in wistar rats exposed to ambient air of Port Har21 court metropolis. The research area is deserved to be of highly interest, and the reseach work is relatively completed. I suggest it published after minor revisions.

  1. The abbr. should be defined at its first appearance, such as PAH in abstract.
  2. Figs. 4-10 look similar, the authors need to conclude the results with some tables or Figure

Author Response

Authors have made all necessary corrections.

Reviewers comment 1:

S/N

Comment

Response/Action

1

The abbr. should be defined at its first appearance, such as PAH in abstract.

Done

2

Figs. 4-10 look similar; the authors need to conclude the results with some tables or Figure

They are not similar; each graph represents a type of PAH compound.

Reviewer 2 Report

IJERPH 1159601

This is a research related with the PAHs concentration found in blood rats exposed to polluted indoor and outdoor ambient air. The results are interesting and deserve to be published. Nevertheless, some issues related with the quality control of chemical analyses. Additionally, authors claim that people living in that city are exposed to high levels of PAH and are in a high risk when they did not do any comparison with other analyses and did not a risk analyses either. They should solve these issues before publication.

Introduction

One of the most important concepts in toxicity studies is that of oxidative stress, which is little discussed in the introduction, with references that only mention it, when there are several studies carried out with rats where oxidative stress is determined. In the rest of the manuscript very little is discussed about oxidative stress and the references are related to studies of particles and PAHs or oxidative stress in cells, rather than oxidative stress determined in studies with rats or mice. I suggest to complement with at least the following references:

  • Danielsen PH, Loft S, Jacobsen NR, et al. Oxidative stress, inflammation, and DNA damage in rats after intratrachealinstillation or oral exposure to ambient air and wood smoke particulate matter. Toxicol Sci 2010; 118(2): 574–585.
  • OG Aztatzi-Aguilar, A Valdés-Arzate, Y Debray-García, ES Calderón-Aranda, M Uribe-Ramirez, L Acosta-Saavedra, ME Gonsebatt, JA Maciel-Ruiz, P Petrosyan, V Mugica-Alvarez, MC Gutiérrez-Ruiz, LE Gómez-Quiroz, A Osornio-Vargas, J Froines, MT Kleinman, and A De Vizcaya-Ruiz. Exposure to ambient particulate matter induces oxidative stress in lung and aorta in a size- and time-dependent manner in rats. Toxicology research and application. 2:1-15.

Methodology

  • PAHs are usually determined with mass gases or HPLC since optimizing the chromatographic separation is critical for PAH analysis due to the isobaric compounds that commonly coelute making quantitation difficult. How was the control assurance of PAH analysis? How were constructed the calibration curves?, did the authors carried out recovery analysis in order to ensure an adequate quantification? Please include an example of a chromatogram.
  • How was measured the PM concentrations indoor and outdoor.
  • Pleas check the PAH names in the whole document since there are several mistakes.

Results

The report of PM levels is too vague: “ranges of 32 μg/m3 to 467 μg/m3“, the means and medians should be mentioned also. Please include a short Table with the daily concentrations of indoor and outdoor PM2.5 and PM10, during the experiment days. How many days the PM concentrations exceeded the standards?

Line 361. With the results of this study authors cannot suggest that “people in non-hydrocarbon and petroleum-free environments are at lower risk of cancers and general tissue toxicity compared to residents inhaling air polluted with PAHs bonded particulate matter due to petroleum and hydrocarbon related industrial activities”.  Their study is only comparing outdoor and indoor PAH levels and indoor were quite low in a city with industrial activities related to petroleum (meaning that people inside is not affected by those activities). They can only make conclusions between indoor and outdoor environments.

In addition, a Table with comparative results of this study with others is missing. In the whole manuscript authors claim that PAH levels are very high, but compared with which? Are their results higher than in other cities? How much? Only with comparisons authors can conclude that inhabitant’s breath high levels of particles and PAHs.

Since the most important results are related with the greater levels of PAHs outdoor than indoor, they could investigate how much time a person stays inside and outside, to talk about risk.

Discussion

I cannot agree with this statement: “Similarly, the result, then they can only also proves that the risk of toxicity, carcinogenicity, and mutagenicity due to high concentrations of PAHs are likely higher in inhabitants living in hydrocarbon industrial polluted environments than those in non-hydrocarbon industrial areas [5,44,54,62,84,102–104]”.

Authors did not a risk analysis study to prove a high toxicity, carcinogenic and mutagenic situation.

Risk analysis include the hours inside and outside and other considerations that were not assessed here.

Conclusions

Again, authors cannot claim: “The findings demonstrate that the PAHs level in the atmosphere of Port Harcourt metropolis is notably high”. If they did not a comparison with other studies and places.

Authors can only conclude about their own results. PAHs indoor and PAHs outdoor. If they want to conclude about high concentrations they should relate their research with standards, or compare with other studies.

Author Response

All observations and queries have been addressed in the review comment document.

Reviewer Comment 2:

S/N

Comment

Response/Action

1

This is a research related with the PAHs concentration found in blood rats exposed to polluted indoor and outdoor ambient air. The results are interesting and deserve to be published. Nevertheless, some issues related with the quality control of chemical analyses. Additionally, authors claim that people living in that city are exposed to high levels of PAH and are in a high risk when they did not do any comparison with other analyses and did not a risk analysis either. They should solve these issues before publication.

This summary of observations and queries are addressed in the different sections where elaborate comments were made.

2

Introduction

One of the most important concepts in toxicity studies is that of oxidative stress, which is little discussed in the introduction, with references that only mention it, when there are several studies carried out with rats where oxidative stress is determined. In the rest of the manuscript very little is discussed about oxidative stress and the references are related to studies of particles and PAHs or oxidative stress in cells, rather than oxidative stress determined in studies with rats or mice. I suggest to complement with at least the following references:

Danielsen PH, Loft S, Jacobsen NR, et al. Oxidative stress, inflammation, and DNA damage in rats after intratrachealinstillation or oral exposure to ambient air and wood smoke particulate matter. Toxicol Sci 2010; 118(2): 574–585.

OG Aztatzi-Aguilar, A Valdés-Arzate, Y Debray-García, ES Calderón-Aranda, M Uribe-Ramirez, L Acosta-Saavedra, ME Gonsebatt, JA Maciel-Ruiz, P Petrosyan, V Mugica-Alvarez, MC Gutiérrez-Ruiz, LE Gómez-Quiroz, A Osornio-Vargas, J Froines, MT Kleinman, and A De Vizcaya-Ruiz. Exposure to ambient particulate matter induces oxidative stress in lung and aorta in a size- and time-dependent manner in rats. Toxicology research and application. 2:1-15.

Included the articles and summarily presented their findings (ref no. 142 and 143).

Depending on the dose and duration of exposure, in-vivo and in-vitro animal models using rodents and mammal tissues (cultured human and placenta) found that both isolated and or combined PAHs induces and promote skin irritation, and inflammation, carcinogenicity, immune suppression, genotoxicity, and mutagenicity and teratogenicity leading to embryotoxic effects [43–47]. The compounds prompt disease through oxidative stress [142, 143] and change in the genome; thus, altering DNA methylation and expression of the specific gene [5, 48].

3

Methodology

PAHs are usually determined with mass gases or HPLC since optimizing the chromatographic separation is critical for PAH analysis due to the isobaric compounds that commonly coelute making quantitation difficult. How was the control assurance of PAH analysis? How were constructed the calibration curves? did the authors carried out recovery analysis in order to ensure an adequate quantification? Please include an example of a chromatogram.

How was measured the PM concentrations indoor and outdoor.

Please check the PAH names in the whole document since there are several mistakes.

It is difficult to get HPLC within our study area in Nigeria, and when available it is exorbitantly high.

We used a facility that has been accredited by our Department of Petroleum Resources and used by Shell-Total-Elf Joint Venture

Chromatogram along with the excel data was deposited at Harvard Dataverse Repository, available at https://doi.org/10.7910/DVN/4GKFHE

[Line 242 – 263]

Measurement of indoor and outdoor PM concentrations: A Handheld (China Way CW-HAT200) SPM optical meters were used to monitor the daily (at mornings and eve-nings) particulate matter (PM2.5 and PM10), relative humidity, and ambient temperature in the indoor and outdoor air by optical method. Measurement was conducted in the mornings (8am) during exposure of outdoor rats and evenings (5pm) when they are re-turned inside. The indoor rats were always inside and the particulate matter measured. The daily (90 working days) measurements for both indoor and outdoor air were recorded in a dedicated notebook

For recovery analysis, to ensure an adequate quantification, metric spike method was used. The compound was spiked into separately prepared aliquots of the MM5 train condensate samples before analysis. The spiked aliquots are then analyzed, and the spike recovery is calculated. Recovery of these spikes (at ≥80%) provided an independent indicator of method accuracy and quantification relative to the sample matrix, by assuming that the spiking compound have chemical characteristics that are identical to the PAH target compounds.

4

Results

The report of PM levels is too vague: “ranges of 32 μg/m3 to 467 μg/m3“, the means and medians should be mentioned also. Please include a short Table with the daily concentrations of indoor and outdoor PM2.5 and PM10, during the experiment days. How many days the PM concentrations exceeded the standards?

Done, and the tables have been deposited in repository; available at

Line 361. With the results of this study authors cannot suggest that “people in non-hydrocarbon and petroleum-free environments are at lower risk of cancers and general tissue toxicity compared to residents inhaling air polluted with PAHs bonded particulate matter due to petroleum and hydrocarbon related industrial activities”.  Their study is only comparing outdoor and indoor PAH levels and indoor were quite low in a city with industrial activities related to petroleum (meaning that people inside is not affected by those activities). They can only make conclusions between indoor and outdoor environments.

We removed the statement on risk. As we understood that a risk analysis would be required to justify the statement.

The study compared it with Baseline data obtained from an unpolluted environment in Enugu.

This has been taken care of in the last paragraph of the introduction “To this end, this study, using an animal model [57] compared the PAH level concentrations in the blood tissue of Wistar rats exposed to the indoor and outdoor ambient air of a polluted environment in Port Harcourt, and how they deviate from established baseline obtained from non-polluted environment.”

In addition, a Table with comparative results of this study with others is missing. In the whole manuscript authors claim that PAH levels are very high, but compared with which? Are their results higher than in other cities? How much? Only with comparisons authors can conclude that inhabitant’s breath high levels of particles and PAHs.

We did not intent to portray a comparative risk with a standard (yet)!

What we are looking at the tissue accumulation of known PAHs, to differentially highlight the differences in the indoor and outdoor concentrations, and unpolluted environment (See Fig. 3)

The significance of the problem is not about how high, but the presence of PAHs, and accumulative differences in polluted and unpolluted environment. An important general concept in toxicity and disease risk in safety is the triage of exposure, dose, duration and bioaccumulation.

Since the most important results are related with the greater levels of PAHs outdoor than indoor, they could investigate how much time a person stays inside and outside, to talk about risk.

At this point we should clarify that the use of the term “risk” is not experimentally derived risk index rather the possibility of something bad happening based on the situation of exposure to a hazardous state.

Not the risk analysis model.

5

I cannot agree with this statement: “Similarly, the result, then they can only also proves that the risk of toxicity, carcinogenicity, and mutagenicity due to high concentrations of PAHs are likely higher in inhabitants living in hydrocarbon industrial polluted environments than those in non-hydrocarbon industrial areas [5,44,54,62,84,102–104]”.

Statement has been rephrased to “Several studies found that people living in areas with high concentrations of PAHs such as hydrocarbon industrial polluted environments are at higher risk of toxicity, carcinogenicity, and mutagenicity [5,44,54,62,84,102–104] than those in non-hydrocarbon industrial areas such as Enugu [144].”

Authors did not a risk analysis study to prove a high toxicity, carcinogenic and mutagenic situation.

We were referring to the possibility of something bad happening based on the situation of exposure to a hazardous state based on previous implicating literatures. Nevertheless, the statements are clarified in the reviewed documents.

Risk analysis include the hours inside and outside and other considerations that were not assessed here.

Not done and statements clarified

6

Conclusion

Again, authors cannot claim: “The findings demonstrate that the PAHs level in the atmosphere of Port Harcourt metropolis is notably high”. If they did not a comparison with other studies and places.

OK. Noted.

But we had baseline animal results from Enugu which was compared against the indoor and outdoor results.

We will state that based on the experimental model re

Authors can only conclude about their own results. PAHs indoor and PAHs outdoor. If they want to conclude about high concentrations

they should relate their research with standards, or compare with other studies.

Noted and corrected.

Round 2

Reviewer 2 Report

In general, all the comments were attended and the manuscript was improved